# Comprehensive Comparison of Clinically Relevant Grain Proteins in Modern and Traditional Bread Wheat Cultivars

**DOI:** 10.3390/ijms21103445

**Published:** 2020-05-13

**Authors:** Olha Lakhneko, Maksym Danchenko, Bogdan Morgun, Andrej Kováč, Petra Majerová, Ľudovit Škultéty

**Affiliations:** 1Institute of Virology, Biomedical Research Center, Slovak Academy of Sciences, Dubravska 9, 84505 Bratislava, Slovak Republic; olakhneko@icbge.org.ua (O.L.); viruludo@savba.sk (Ľ.Š.); 2Institute of Cell Biology and Genetic Engineering, National Academy of Sciences of Ukraine, Akademika Zabolotnoho 148, 03143 Kyiv, Ukraine; bmorgun@icbge.org.ua; 3Institute of Plant Genetics and Biotechnology, Plant Science and Biodiversity Center, Slovak Academy of Sciences, Akademicka 2, 95007 Nitra, Slovak Republic; 4Institute of Neuroimmunology, Slovak Academy of Sciences, Dubravska 9, 84510 Bratislava, Slovak Republic; andrej.kovac@savba.sk (A.K.); petra.majerova@savba.sk (P.M.); 5Institute of Microbiology, Czech Academy of Sciences, Videnska 1083, 14220 Prague, Czech Republic

**Keywords:** *Triticum aestivum* L., food quality, cereal allergens, discovery proteomics, gluten, celiac disease

## Abstract

Bread wheat (*Triticum aestivum* L.) is one of the most valuable cereal crops for human consumption. Its grain storage proteins define bread quality, though they may cause food intolerances or allergies in susceptible individuals. Herein, we discovered a diversity of grain proteins in three Ukrainian wheat cultivars: Sotnytsia, Panna (both modern selection), and Ukrainka (landrace). Firstly, proteins were isolated with a detergent-containing buffer that allowed extraction of various groups of storage proteins (glutenins, gliadins, globulins, and albumins); secondly, the proteome was profiled by the two-dimensional gel electrophoresis. Using multi-enzymatic digestion, we identified 49 differentially accumulated proteins. Parallel ultrahigh-performance liquid chromatography separation followed by direct mass spectrometry quantification complemented the results. Principal component analysis confirmed that differences among genotypes were a major source of variation. Non-gluten fraction better discriminated bread wheat cultivars. Various accumulation of clinically relevant plant proteins highlighted one of the modern genotypes as a promising donor for the breeding of hypoallergenic cereals.

## 1. Introduction

Bread wheat (*Triticum aestivum* L.) is a valuable cereal widely used in the human diet or livestock feed, and the dominant crop in temperate countries. It is an essential source of nutrients and other beneficial components. World production of this crop reaches 725 million tons annually, which is 30% of all harvested cereals (http://www.fao.org/3/a-I8080e.pdf). Altogether, more wheat proteins are consumed by humanity than from any other plant or animal. This crop is traditionally vital for European nations, though it has broad geographic distribution. The success of wheat largely depends on its adaptability to a wide range of environments, high yield potential, and relevance to the human culture [1].

The wheat grain contains about 16% of proteins, which are classified according to their solubility: In water—albumins, in salt—globulins, in alcohol—gliadins, or in alkali—glutenins. Typically, wheat flour proteome consists of 35% glutenins, 45% gliadins, and only 20% other proteins. Glutenins and gliadins are related and defined as gluten; multiple genes encode them at complex loci [2]. Glutenin fraction represents a complex polymer, stabilized by inter-chain disulfide bonds. Glutenins are classified into high molecular weight (HMW) and low molecular weight (LMW) subunits [2]. A combination of different HMW alleles of x- and y-type subunits defines the elasticity and strength of the dough [3]. Likewise, LMW subunits are determinants of dough extensibility in bread wheat [4]. However, the exact role of each specific LMW glutenins remains largely mysterious. For instance, Lee group found that a single genetic locus played only a minor role in quality variation, although it was the most diverse [5]. Monomeric gliadins are another dominant part of storage proteins. They are divided into α/β-, γ-, and ω-classes according to differences in the primary structure and the number of conserved cysteine residues [6,7]. Gliadin genetic regions are characterized by the complex structure and may cover over 50 alleles, a lot of which are actually expressed, but also a number of them are pseudogenes [8,9]. Gliadins contribute to bread-making quality through covalent and non-covalent bonds with other polymeric gluten components, forming the fine gluten film network and improving gas retention, viscosity, and cohesiveness of dough. Some studies demonstrated the importance of the balance between glutenin and gliadin fractions for boosting bread-making quality [10,11]. Globulins and albumins, collectively referred as metabolic proteins, compose a minor part of grain proteome. They are marginally connected to the technological quality by defining milling properties, but are indispensable for the plant physiology [6].

Modern plant breeding has led to the development of multiple wheat cultivars with superior bread-making quality. Albeit, storage proteins can cause food intolerance or allergy in susceptible individuals. People are exposed to wheat-derived products through ingestion, inhalation, or skin contact. Wheat sensitivities are classified in autoimmune conditions (having T-cell or IgA nature): Celiac disease, gluten ataxia, gluten neuropathy, dermatitis herpetiformis; and allergic disorders (mediated by IgE): Respiratory allergy, food allergy, wheat-dependent exercise-induced anaphylaxis, contact urticaria [12,13,14]. Etiology of wheat intolerances grounds in inefficient digestion of the consumed gluten-containing food. This may happen because glutenins and gliadins are enriched with glutamine and proline, leading to restricted cleavage by gastric enzymes [14]. Notably, a thorough study reported considerable variation in the T-cell responses of 14 celiac patients, indicating the existence of numerous active epitopes [15]. Proteomics greatly contributed to the understanding of allergy and intolerance to wheat products, through qualitative and structural characterization of the allergenic and toxic peptides [16]. Of note, researchers proved that besides gluten, metabolic proteins are also of medical concern. Celiac disease patients showed antibody reactivity to non-gluten proteins: Serpins (the most frequently), purinins, α-amylase/protease inhibitors, globulins, and farinins. Recombinant proteins confirmed a robust humoral immune response [17,18].

Genetic and environmental factors affect the technological properties of wheat in a rather unpredictable way. One route for safe food is biotechnological creation of transgenic lines; another option is through exploiting rich traditional genetic resources to lower the amount of harmful epitopes [8]. There is a serious public perception issue with genetically modified organisms, yet it has no reliable scientific arguments. An effective approach to reduce allergenicity/toxicity is the silencing of target genes, like ω-gliadins. However, thorough proteomic evaluation of transgenic lines before mass production is recommended because even the same construct can have different effects on grain proteome [19]. Piston group reported that suppressing γ-gliadins resulted in counterbalance of ω- and α/β-classes [20]. On the other hand, despite its recent origin, there is immense diversity among *T. aestivum* cultivars. Experts estimated the existence of at least 25,000 genotypes. A comprehensive study, using immunologic assays, demonstrated a large variation in the amount of reactive peptides among wheat cultivars [21]. The authors concluded that sufficient genetic variation enables the selection of cereals with reduced risk of causing disorders. Such genotypes would allow patients to enjoy a more balanced diet. Intuitively, landraces were believed to synthesize lower amounts of clinically relevant proteins. However, a recent study contradicted this theory [22] and the Simsek group failed to discover any association between release year of wheat cultivars and quantity of immunogenic epitopes in α-gliadins [23]. Finally, it was suggested that more thorough investigations are necessary to clarify potential lower toxicity of old landraces because available data are highly heterogeneous and quite often, growth conditions are unaccounted [24]. Differences in metabolic proteins were shown to be more informative for the discrimination of wheat cultivars [25,26].

Proteomics allows a realistic estimation of the amount of specific grain storage proteins, overcoming the limitations of molecular tools detecting genes/transcripts. Accurate quantification of grain proteins, particularly gluten, is far from a trivial task. The first challenge is the extraction of physicochemically diverse subgroups of polypeptides. Often, a sequential extraction protocol is used, but this introduces inherent analytical variation [27]. Anionic detergent SDS-assisted extraction seems superior, allowing to obtain a more complex mixture in a single step [19]. Wheat grain proteins have high sequence homology and content of hydrophobic amino acids; moreover, immunogenic/toxic peptides are usually resistant to proteolysis. Gluten proteins are rich in proline/glutamine; thus, thorough precise analysis requires alternative proteases [28]. Gel-based methods remain commonly used due to the visualization of multiple isoforms. Long stretches of repetitive sequences complicate unique peptide assignment when using mass spectrometry. Nevertheless, ultrahigh-performance liquid chromatography (UHPLC) with targeted mass spectrometry offers accurate multiplex quantification options, emerging as a viable alternative to standard enzyme-linked immunosorbent assays for allergenicity/toxicity assessment of food products [29,30,31].

Herein, a wide-range of bread wheat proteins was extracted by SDS-containing buffer and separated using two-dimensional gel electrophoresis (2-DE). Then, we identified differentially abundant grain proteins revealing polymorphism among cultivars. UHPLC of chymotryptically digested extracts with ion mobility-enhanced direct mass spectrometry quantification complemented the dataset. Finally, using bioinformatics, we assessed the clinical relevance of variably accumulated proteins.

## 2. Results and Discussion

### 2.1. Separation of Grain Proteins by Two-Dimensional Gel Electrophoresis

Wide-range IPG strips (3–10) covered diverse proteins accumulated in wheat grain. Triplicate gel images are presented in the Appendix A, confirming the excellent quality and reproducibility of the analysis (Appendix A). Detected spots had the pI range 4.0–9.5 and molecular mass interval 11–129 kDa. This analytical approach detected 810 protein spots among two modern and one historical genotype. Strict statistics and effect size criteria revealed 66 differently abundant gel spots annotated on the reference image (Figure 1). The quantitative values are presented in the Appendix A as means and standard deviations of three biological replicates, together with identification details for differentially abundant proteins (Appendix A). Notably, unsupervised statistics—principal component analysis—confirmed the clustering of differentially abundant gel spots according to the genotype (Figure 2A). Moreover, it suggested that cultivar Sotnytsia is distant relative to the others. In order to identify differentially accumulated wheat grain proteins, we applied parallel in-gel digestion with multiple enzymes and identified 49 gel spots (Table 1) with a sufficient number of matched peptides (Appendix A). Major gluten and non-gluten fractions were well represented: 8 glutenins, 18 gliadins, and 23 metabolic proteins.

Using comprehensive databases, we annotated 22 protein spots as allergenic or toxic. Importantly, all differentially accumulated glutenin subunits are causing disorders. The first group includes food allergen Tri a 26 [32]. HMW-PW212 glutenin (spot 390) was over 10 times more abundant in Sotnytsia compared to Ukrainka and Panna. HMW-12 glutenin (*HMW-GS DY*) was identified in two gel spots 1066 and 4114. The former accumulated in Sotnytsia compared to Ukrainka, and the latter showed higher content in Ukrainka than in Panna. It is noteworthy that these spots had a very distinct isoelectric point (Figure 1). We identified HMW-DX5 glutenin (*Glu-1D-1D*) in four gel spots: (i) 4045 and 4332 accumulated in Panna compared to Sotnytsia, (ii) 4345 was more abundant in Ukrainka versus Sotnytsia, and (iii) 4351 showed a higher amount in both Panna and Ukrainka respective to Sotnytsia (Table 1). The second group, comprising food allergen Tri a 36 [32], included only LMW-B2 glutenin (*Glu-B3-2*) in a single gel spot 4126 that was more abundant in Panna compared to Ukrainka. This protein showed a different trend in LC-MS based quantification.

The vast majority of differentially accumulated medically relevant gliadins belong to the food allergen Tri a 20 class. This group unites γ-gliadins causing wheat-dependent exercise-induced anaphylaxis [33], perhaps because of homology to ω-gliadins. We detected four different γ-gliadins, each in multiple gel spots. The γ-gliadin P21292 was more abundant in Panna relative to Sotnytsia and Ukrainka in 4179 and more than 5-fold in 1803. The γ-gliadin D4 (*Gli1*) was more abundant in Sotnytsia compared to other genotypes (2161) or only to Ukrainka (2078). The γ-gliadin D3 (*Gli1*) from 1896 accumulated in Sotnytsia relative to Ukrainka, and from 1961 was more abundant in landrace compared to modern bread wheat cultivars. The former is likely a minor post-translational modification (PTM) with a shifted isoelectric point of the latter, which is one order more abundant in all samples. We detected γ-gliadin A1 (*Gli1*) in three gel spots 1649, 1656, and 4220 accumulated more than five times in Ukrainka compared to Panna and Sotnytsia (Table 1). The same pattern characterized α/β-gliadin I0IT62 (spot 3866) not associated with any Allergome identifier. Another α/β-gliadin AII, without this database annotation, accumulated in both modern cultivars compared to landrace (spot 1669) or only in Sotnytsia versus Ukrainka (spot 4403). Gliadins identified in a smaller proportion of differentially accumulated gel spots do not have an association with any human disorder. A protein belonging to the UniRef cluster, which includes avenin-like proteins, was differentially abundant in multiple gel spots 2645, 2689, 4169, 4481, and 4482. The majority of them accumulated in Panna and Ukrainka compared to Sotnytsia, particularly 2689 more than 5-fold. All these spots shared the same area on the gel, but one 4169, having slightly lower pI and opposite accumulation (Figure 1). A protein similar to δ-gliadin D1 (spot 4491) accumulated over 5-fold in landrace relative to newer cultivars.

In the non-gluten grain protein fraction, we discovered a trypsin/α-amylase inhibitor CMX2, which belongs to the Tri a CMX group, and predominantly causes a non-celiac gluten sensitivity [18,34]. This protein from spot 3607 accumulated in Panna compared to Sotnytsia and Ukrainka. Serpin-Z1A (*WZCI*) in spot 1581 was more abundant in Ukrainka compared to Sotnytsia. It belongs to the Tri a 33 group and elicits a broad spectrum of sensitizations [35]. The largest proportion of differentially abundant gel spots comprised metabolic proteins, which are neither allergens nor toxins. We identified several storage proteins as globulin-3A (*Glo-3A*) accumulated in Sotnytsia compared to Panna (spot 4155), or vice versa (spot 1051), which showed very distinct pI. A protein similar to globulin-1 S allele (spot 4055) accumulated more than 5-fold in Sotnytsia compared to other genotypes. Protein similar to globulin 3 (spot 4272) was more abundant in Ukrainka versus Panna. Cupin domain protein in spot 2719 (also revealed in complementary LC-MS analysis) accumulated in Panna and Ukrainka compared to Sotnytsia. Among the proteins involved in primary metabolism was similar to β-amylase (spots 1082 and 1084) more abundant in Panna versus Sotnytsia and Sotnytsia versus Ukrainka, respectively. The former seems to be a minor modification of the latter. Cytoplasmic glycolytic enzyme pyrophosphate—fructose 6-phosphate 1-phosphotransferase subunit β (*PFP-β*) from spot 969 accumulated in Ukrainka compared to Sotnytsia. The enzyme involved in the biosynthesis of saccharides (spot 1169) was more abundant in Ukrainka versus Panna. Alanine-tRNA ligase (spot 3358) accumulated in Sotnytsia versus Panna and methyltransferase in spot 4404 showed a higher amount in Sotnytsia compared to Ukrainka. Besides, we discovered several stress proteins, like dehydrin (spot 1025) or a protein similar to dehydroascorbate reductase (*DHAR*) in spot 2335 that more than 5-fold accumulated in landrace compared to the newer genotypes or in Sotnytsia versus Panna and Ukrainka, respectively. A protein similar to serpin-N3.2 (spot 2386) was more abundant in Sotnytsia compared to Ukrainka. The last seven spots, which varied across investigated bread wheat genotypes, are not annotated.

### 2.2. Label-Free Quantification Complemented the Comparison Between Bread Wheat Cultivars

To complement the gel-based dataset, we employed a comprehensive UHPLC profiling of peptides, followed by ion mobility-enhanced label-free quantification using an accurate peak area principle. Ion mobility allows another separation dimension, complementing reverse phase chromatography. As a result, we reproducibly quantified 127 proteins among samples. We presented the values as means and standard deviations of three biological replicates together with identification and annotation details in the Appendix A. The whole list of identified peptides is a valuable reference for the future development of targeted quantification methods (Appendix A). Criteria for selecting differentially abundant proteins were the same as with gel-based analysis in terms of effect size and statistical significance for consistency. Thereby, 12 proteins satisfied these parameters (Table 2). Of note, the principal component analysis indicated sample clustering according to the genotype and closer relation between cultivars Panna and Ukrainka, concordant to 2-DE (Figure 2B). Principal fractions were six glutenins, two gliadins, and four metabolic proteins.

Seven of the discovered differentially abundant proteins are known to contain allergenic/toxic epitopes according to the database sources. As in the case of 2-DE quantification, all variable glutenin subunits can affect human health. They belong to the Tri a 36 group [32], according to Allergome classification. Glutenin LMW-D1 (*Glu-D3-3*) and glutenin LMW-1D1 (*Glu-D3-2*) showed higher abundance in Ukrainka compared to Sotnytsia. LMW glutenin subunit (*Glu-A3-16*), LMW glutenin subunit (*Glu-B3-1*), and glutenin LMW-A2 (*Glu-A3-11*, over 5-fold) were more abundant in the grain of cultivars Panna and Ukrainka versus Sotnytsia. Glutenin LMW-B2 (*Glu-B3-2*) was more than 10-fold abundant in Panna and Ukrainka respective to Sotnytsia. Single toxic protein from the gliadin group, belonging to the food allergen class Tri a 20 [33], similar to γ-gliadin accumulated in Ukrainka contrasted to Sotnytsia. Another gliadin related protein, avenin-like a6, was more abundant in Ukrainka compared to Panna. Avenins are not directly integrated into the gluten polymer through disulfide bonds unless incorporated by reduction and reoxidation during dough making [36]. Among variable proteins belonging to a non-gluten group, we discovered an uncharacterized membrane protein having a dentin matrix domain as more abundant in Sotnytsia compared to other cultivars. The thaumatin family (pathogenesis-related group five) protein accumulated in Sotnytsia relative to Ukrainka. A storage protein having a cupin domain A0A3B5XV32 was more abundant in Panna compared only to Sotnytsia, and another cupin domain protein A0A1D5Y5R1 accumulated in Sotnytsia and Ukrainka versus Panna.

Contrasting differentially abundant proteins from complementary approaches, we revealed only two common accessions and just one of them—LMW-B2 glutenin (*Glu-B3-2*)—is a known food allergen. In 2-DE, we detected about 2-fold higher amount of this protein in Panna (significantly) and Sotnytsia (below effect size threshold) relative to Ukrainka, but somehow contradictory, the LC-MS approach pointed to more than 10-fold higher abundance in Panna and Ukrainka compared to Sotnytsia. Likely, this discrepancy reflects the specific characteristics of the particular analytical method. On the other hand, storage protein with cupin domain protein accumulated in Panna and Ukrainka compared to Sotnytsia according to both 2-DE evaluation, and direct mass spectrometry quantification (below effect size threshold for Ukrainka).

A small overlap between LC-MS and 2-DE proteomic datasets is puzzling. However, such phenomenon is rather common [37]. We assume that some discrepancy is explained by PTMs, which were separated on gels, but not distinguished by direct LC-MS. Such as HMW-12 glutenin differentially accumulated in two gel spots (Table 1) but showed the same amount among cultivars according to the label-free quantification (Appendix A). On the other hand, some proteins highlighted as varied in the LC-MS dataset, for example, LMW-A2 glutenin (Table 2), could be among a few protein spots, which we failed to identify.

### 2.3. Effective Workflow to Overcome the Analytical Challenges of Grain Proteome Profiling

Proteomic tools have been exhaustively used to evaluate the genetic diversity of wheat germplasm from different continents at the level of allelic polymorphism of glutenins and gliadins—the two main components of gluten. More recently, proteomics became fundamental to understand the impact of specific gluten proteins on wheat quality [38]. We used anionic detergent SDS to extract physicochemically diverse subgroups of wheat storage proteins [19]. Consequently, we obtained a complex mixture with both gluten and other proteins in a single step for better analytical reproducibility. Since gluten proteins are rich in proline/glutamine, alternative proteases might be necessary for efficient analysis. Multi-enzymatic digestion is preferable to generate unique peptides because thermolytic, chymotryptic, and tryptic peptides match different parts of protein sequences [28]. Considering the fact that more hits could be identified after chymotryptic digestion of gluten proteins than using other enzymes [39], we used chymotrypsin as the first choice protease, even though scoring algorithms are optimized for classic trypsin digestion. Our data confirmed that processing of gel spots, unassigned after chymotryptic digestion, with alternative enzymes, added several more hits to the cumulative list.

The comparative advantage of 2-DE is a visualization of multiple isoforms. Nevertheless, direct mass spectrometry quantification is gaining momentum as a preferred analytical tool for the discovery proteomics since it is fast, robust, deep, and with mature bioinformatic algorithms for reliable quantification. However, peptide centered analysis—LC-MS—has intrinsic limitations, such as the loss of intact protein information, consequently the inability to grasp combinations of PTMs. Repetitive sequence motifs, frequent in gluten proteins, enormously complicate unique peptide assignment with mass spectrometry. Therefore, using data-independent mass spectrometry, Hajduch laboratory quantified only 34 gliadins and 22 glutenins in wheat grain, concluding that their approach is a reproducible quantitative method for the determination of gluten protein content in the highly complex matrix [40]. Quadrupole time-of-flight mass spectrometer with ion mobility worked very well for sequencing longer peptides, crucial to distinguish individual gluten proteins [41]. Using the same type of instrument, we also confirmed multiple glutamine deamidations, as reported earlier [41]. Nevertheless, due to the inherent challenges, like the dominance of only a few related (sharing long sequence stretches) storage protein classes, direct mass spectrometry in the discovery mode was not so effective. Despite effort into developing an optimal analytical workflow, we quantified a modest number of proteins.

### 2.4. Both Gluten and Non-Gluten Proteins Contain Allergenic or Toxic Motifs

Consolidating both 2-DE and label-free datasets, we revealed that the majority of proteins showing variable amounts in studied genotypes belong to a metabolic group—27 hits, 44%; the next are gliadins—20 proteins, 33%; and the least glutenins—14 hits, 23% (Figure 3). Contrastingly, among allergenic/toxic proteins, the highest diversity showed glutenins with 14 proteins, 48%; gliadins included 13 hits, 45%; and the non-gluten group consisted of only two proteins, 7%. Traditional landrace Ukrainka showed the highest cumulative coefficient for proteins with allergenic/toxic epitopes (Figure 4). Next, we extracted information on known epitopes in sequences of discovered allergenic/toxic proteins (Table 3). The number, as well as density (number of toxic sequences divided by the quantity of all amino acids) of unique celiac motifs according to GluPro and the number of medically relevant epitopes according to ProPepper, characterized the detected polypeptides. The highest number of such epitopes 127 occurred in α/β-gliadin AII, yet the highest density of the toxic celiac motifs occurred in γ-gliadin P21292; the former was particularly notable in Sotnytsia, accumulating in two gel spots, the latter also in two gel spots was more abundant in Panna. Contrary, HMW-12 glutenin (*HMW-GS DY*), differentially abundant in two gel spots, had only six epitopes and minimal density of unique celiac motifs.

Herein, we evaluated the variability of all proteome fractions, including metabolic proteins and globulins, because non-gluten proteins better discriminated wheat cultivars and lines [25,26,42]. Structure modeling and epitope prediction confirmed the presence of linear homologs of celiac disease epitopes in seed storage globulins, highlighting that reactive response may be developed not exclusively to prolamins [43]. We confirmed non-gluten proteins as the primary variable group (Figure 3), yet discovered little diversity among allergens/toxins within metabolic proteins (Table 3). Recently, researchers analyzed the composition of epitopes relevant to celiac disease and wheat-dependent exercise-induced anaphylaxis in the gliadins from the flour of cultivar Keumkang using 2-DE. They confirmed a strong immunogenic potential of specific α/β- and γ-gliadins [44]. Our study highlighted five unique γ-gliadins and two α/β-gliadins among bread wheat genotypes (Table 3). High-resolution 2-DE discovered nine subunits of LMW glutenins as predominant IgE-binding antigens causing food allergy [45]. In line with this report, our analysis also showed multiple potentially toxic glutenins differentially accumulated among Ukrainian bread wheat cultivars (Table 1 and Table 2).

Multiple databases contain useful curated and annotated information about the role of wheat proteins in different pathologies. However, the heterogeneity of these databases prevents direct connectivity. Developers of one useful resource Allergome have the ambition to make it a common platform where experimental and clinical data will be merged [46]. ProPepper indexes linear epitopes with proven T- or B-cell specific activity [47] and GluPro database associates sequences of gluten proteins with celiac disease by annotating number and density of unique medically relevant motifs [48]. We received largely correlated indices from ProPepper and GluPro analysis of clinically relevant proteins in *T. aestivum* genotypes (Table 3).

### 2.5. Modern Breeding, Traditional Landraces, and Biotechnology for Safe Wheat

Wheat breeding primarily targeted specific traits such as high yield, disease resistance, and drought tolerance. There is sufficient genetic variation to breed wheat with reduced toxicity, which is beneficial for people with sensitivities [21]. Using accurate targeted mass spectrometry, researchers showed that a modern *T. aestivum* cultivar, Toronto, contained the highest amounts of immunogenic celiac peptides compared with the older one, Minaret, and the tetraploid wheat cultivar [49]. Contrary, a large screening study rebutted the idea that landraces synthesize a lower amount of clinically relevant proteins, showing no association between the age of cultivars and immunogenicity of gliadins [23]. Standardized growth conditions should clarify the genetic potential of lower toxicity for older landraces [24]. In our experiment, grains were harvested from neighboring plots agronomically treated in the same manner, avoiding heterogeneity of growth conditions. Another survey detected immunogenic epitopes having an extremely high quantitative range in both historical and modern spring wheat genotypes. Therein, researchers proposed cultivar Russ with the least amount of immunogenic epitopes, as a suitable starting point for breeding safer wheat for celiac disease patients [23]. Our study also highlighted one of the modern cultivars of Ukrainian selection Sotnytsia, as the genotype with the lowest accumulation of allergenic/toxic proteins in grain (Figure 4). The Igrejas group showed that *T. aestivum* landraces presented higher amounts of immunostimulatory epitopes and concluded that there is a good potential for selection of cultivars with a low content of toxic epitopes via conventional breeding practices [22].

A significant positive correlation was found between release year of cultivars and dough quality characteristics, which could be associated with quantitative variations in glutenin polymeric proteins, and certain subfractions of ω-gliadins [50]. Farmers increasingly replace landraces with modern cultivars, which are less resilient to pests, diseases, and abiotic stresses. Thereby, a valuable source of germplasm may be lost for meeting the future needs of sustainable agriculture in the context of climate change. Landraces are a proven indispensable source of resistance to pathogens, which should be preserved and exploited [51]. Superior stress resilience is due to the natural heterogeneity of the landraces in contrast to modern, more homogeneous cultivars [52].

An alternative approach to make wheat products safer is using powerful genetic engineering. A large survey demonstrated that a wide variation exists in the amount of allergenic polypeptides among wheat cultivars, and the differences detected between genetically engineered lines are within the range of conventional cultivars. Advanced statistics showed that patient sera are yet another key variable [53]. Recently, Barro laboratory reported the first application of genome editing for subtracting immunodominant celiac peptides. Up to 75% of different α/β-gliadin genes were mutated in engineered lines, while immunoreactivity of grain extracts was dramatically reduced. Thus, the authors speculated that such ‘transgene-free’ wheat genotypes could be used to produce less toxic food products and serve as source material to incorporate new traits into elite wheat cultivars [54]. Currently, the most frequent biotechnological approach—RNA interference—was used to produce low-gliadin wheat lines. Separate analyses of gliadins, glutenin subunits, metabolic, and chloroform/methanol-like proteins by a classical 2-DE allowed thorough safety evaluation of the transgenes. Researchers discovered a compensation of total protein, as those lines showed significant accumulation of HMW glutenins, and non-gluten albumins and globulins [27].

## 3. Materials and Methods

### 3.1. Plant Material and Protein Extraction

Three Ukrainian winter wheat cultivars: Sotnytsia, Panna (both modern selection), and Ukrainka (archaic landrace) were selected for this study, being either essential for the contemporary agriculture or older cultivars not widely used in mass production. We hypothesized that they accumulate a different amount of clinically relevant proteins. The crop was grown in the experimental field using standard agrotechnical practice with a random plot design in the Kyiv region. Grain was harvested in the 2017 season. No considerable environmental stress (drought, cold, or pathogens) was recorded during bread wheat growth. Experimental plots were black soil with the appropriate addition of mineral fertilizers.

Panna and Sotnytsia are modern Ukrainian cultivars promising for agricultural mass production. Wheat cultivar Panna, with a unique protein gluten complex, was registered at the beginning of the XXI century. The protein content in its grain is approximately 15.3%, and the technological quality of the flour is above average. Cultivar Panna carries the glutenin allele *Glu-B1-5*, which provides good baking properties, positively influencing elasticity and viscosity of the dough. However, because of lower productivity and the propensity to lay in unfavorable conditions, Panna is not widely planted for grain production. Nevertheless, it is included in breeding programs as a genetic source for improving the quality of winter wheat. On the other hand, cultivar Sotnytsia has a high genetic potential for productivity; moreover, it has superior technological properties for milling and bread making. This cultivar tolerates major pathogens and drought well. Notably, the average productivity of Sotnytsia is about 10% higher than the Ukrainian national standard [55]. Breeding for maximum yield, especially in grain crops, can cause deterioration of protein content. Thus, the modern cultivars of wheat are plausibly inferior to the landrace Ukrainka, comparing the amount and composition of grain storage proteins, particularly gluten. Genotype Ukrainka was developed a century ago, showing decent yield and excellent bread-making qualities. For dozens of years, it was widely grown and even had a reputation of world quality standard. Genetic heterogeneity of glutenin and gliadin genetic loci in historical landrace Ukrainka was reported previously [52,56].

All reagents and solvents of the highest available analytical grade were purchased from Sigma-Aldrich or Merck Millipore, respectively, unless stated otherwise. Grain proteins were extracted according to the developed protocol [57]. Following grinding of 1 g of seeds to fine flour with liquid nitrogen in a mortar, 10 mL of extraction buffer (2% SDS, 10% glycerol, 50 mM dithiothreitol, and 50 mM Tris pH 6.8) was added. The mixture was incubated with vigorous shaking for 1 h, followed by centrifugation at 10,000× *g* for 15 min. Proteins were precipitated from the supernatant with four volumes of cold acetone and kept overnight at −20 °C. The precipitate was collected by centrifugation at 4000× *g*, 4 °C for 15 min, followed by a single acetone wash. The protein precipitate was dried under vacuum and stored at −80 °C.

### 3.2. Two-Dimensional Gel Electrophoresis

Protein extracts from different cultivars were resuspended in the solubilization buffer (8 M urea, 2 M thiourea, 2% CHAPS, and 2% Triton X-100). Then, concentration was measured by Pierce detergent compatible Bradford assay (ThermoFisher Scientific). Protein aliquots of 500 μg were adjusted to 340 μL with isoelectric focusing buffer (8 M urea, 2 M thiourea, 2% CHAPS, 1% amidosulfobetaine-14, 1% Triton X-100, 2% ampholytes, and 1% DeStreak). Upon centrifugation, the supernatant was transferred to 18 cm immobilized pH 3-10 non-linear gradient strips (GE Healthcare). Following overnight passive rehydration, the first dimension separation, according to the isoelectric points, was done in the Ettan IPGphor 3 unit (GE Healthcare). After isoelectric focusing, strips were reduced in 4 mL of equilibration buffer (EQB; 0.1 M Tris pH 6.8, 30% glycerol, 6 M urea, and 3% SDS) with 2% dithiothreitol for 15 min, alkylated in 4 mL of EQB with 2.5% iodoacetamide in the dark for 15 min, and washed in 4 mL of running buffer (25 mM Tris, 192 mM glycine, and 0.1% SDS). Next, they were placed on top of 12% polyacrylamide gels and sealed by 0.5% agarose with 0.002% bromophenol blue in running buffer. The second dimension separation, according to molecular weights, was performed in Protean II xi Cell (Bio-Rad).

Gels were stained by sensitive colloidal Coomassie G-250. Images were digitalized with resolution 300 dpi and 16-bit grayscale pixel depth on Umax ImageScanner (GE Healthcare). Quantitative software-assisted gel analysis was done in SameSpots 5.1 (TotalLab) to reveal protein spots differentially accumulated between cultivars. Firstly, software aligned the images, then dust/stain particles, obvious streaks, and damaged gel areas were filtered. Relative volumes were normalized to the median distribution of reference gel to compensate for minor differences in sample loading. All spots were reviewed and manually edited if necessary. Before statistical analysis, normalized spot volumes were transformed using the inverse hyperbolic sine function. Differentially accumulated proteins were chosen based on ANOVA *p* ≤ 0.01 and ratio ≥ 2.5. Additionally, we applied post hoc Tukey’s honestly significant difference test to assess changes between specific genotypes.

### 3.3. In-Gel Digestion and Filter-Aided Sample Preparation

Differentially abundant protein spots were excised from the gels, washed with 300 µL of 50 mM ammonium bicarbonate in 50% acetonitrile, and dehydrated with 300 µL acetonitrile. Proteins were reduced with 100 µL of 10 mM dithiothreitol in 100 mM ammonium bicarbonate at 50 °C for 30 min. Alkylation was performed with 100 µL of 50 mM iodoacetamide in 100 mM ammonium bicarbonate in the dark for 30 min. Protein spots were digested with 20 µL of 10 ng/µL enzyme (Promega) in 10 mM ammonium bicarbonate and 10% acetonitrile at 25 °C (chymotrypsin) or 37 °C (trypsin), in 50 mM Tris pH 8.0 and 0.5 mM CaCl2 at 60 °C (thermolysin). Peptides were extracted twice with 50 µL of 70% acetonitrile and 1% trifluoroacetic acid.

Total protein extracts were digested according to the filter-aided sample preparation (FASP) protocol [58]. Following activation of centrifugal filter units Microcon Ultracel YM-10 (Merck Millipore) by 200 µL of 1% formic acid and centrifugation at 10,000× *g* for 40 min, 100 µg protein aliquots, adjusted to 200 μL with urea buffer (8 M urea and 100 mM Tris pH 8.5), were loaded on the filter and centrifuged. Subsequently, proteins were washed with 200 µL of urea buffer. The reduction was performed with 200 µL of 10 mM dithiothreitol in the urea buffer at 50 °C for 15 min. Proteins were alkylated with 200 µL of 50 mM iodoacetamide in urea buffer in the dark for 15 min. The excess of iodoacetamide was quenched by dithiothreitol. Final washing was done with 200 µL of 20 mM ammonium bicarbonate, followed by centrifugation of filter units. Then, collection tubes were exchanged, and 75 µL of chymotrypsin 100 ng/μL in 50 mM ammonium bicarbonate was added to the filter. Digestion was carried out at 25 °C overnight. Afterwards, peptides were collected by centrifugation at 10,000× *g* for 20 min. Another 75 µL of 50 mM ammonium bicarbonate was added on the filter and again collected by centrifugation. Finally, 100 µL of 0.1% trifluoroacetic acid was pipetted to acidify the combined flow-throughs. For peptide purification, after FASP, we used Sep-Pak Light C18 cartridges (Waters). Subsequently, the concentration of the peptides was measured by NanoDrop 2000 spectrophotometer (ThermoFisher Scientific).

### 3.4. Tandem Mass Spectrometry for Protein Identification from Gel Spots

The peptides were analyzed by liquid chromatography-tandem mass spectrometry (LC-MS/MS), using nanoAcquity UHPLC (Waters) and Q-TOF Premier (Waters) as described earlier [40] with minor modifications. Samples were separated by BEH130 C18 analytical column (200 mm length, 75 μm diameter, 1.7 μm particle size), using a fast 20 min gradient of 5%–40% acetonitrile with 0.1% formic acid at a flow rate 300 nL/min. The data were recorded in the MSE mode (parallel high and low energy traces without precursor ion selection) and processed using ProteinLynx Global Server 3.0 (Waters). Spectra were searched against wheat proteome sequences downloaded from UniProt in April 2018 (136,892 entries, uniprot.org). Search parameters were as specified in the following chapter for chymotrypsin, but one allowed miscleavage for trypsin and thermolysin. Thermolysin was defined as cutting on N-terminus after alanine, phenylalanine, isoleucine, leucine, methionine, and valine, but not before proline. Identities were accepted if two or more different peptides with a score higher than 95% reliability threshold were matched. Reliability scores were adjusted based on the distribution of target/decoy queries. In the cases when several sequences matched spectra from a single gel spot, we reported accession with the highest number of reliable peptides.

### 3.5. Relative Quantification by Direct Mass Spectrometry and Bioinformatics

Aliquots of purified complex peptide mixtures of 300 ng were separated in biological triplicate, using Acquity M-Class UHPLC (Waters) as described earlier [59]. Samples were loaded onto the nanoEase Symmetry C18 trap column (20 mm length, 180 μm diameter, 5 μm particles size). After 2 min of desalting/concentration by 1% acetonitrile containing 0.1% formic acid at a flow rate 8 μL/min, peptides were introduced to the nanoEase HSS T3 C18 analytical column (100 mm length, 75 μm diameter, 1.8 μm particle size). For the thorough separation, a 90 min gradient of 5%–35% acetonitrile with 0.1% formic acid was applied at a flow rate of 300 nL/min. The samples were nanosprayed (3.1 kV capillary voltage) to the quadrupole time-of-flight mass spectrometer Synapt G2-Si with ion mobility option (Waters). Spectra were recorded in a data-independent manner in high definition MSE mode. Ions with 50–2000 m/z were detected in both channels, with a 1 s spectral acquisition scan rate.

Spectra were preprocessed with the Compression and Archival Tool 1.0 (Waters) to reduce noise, removing ion counts below 15. Data processing was done in Progenesis QI 4.0 (Waters) using the workflow outlined in the literature [59]. For peak picking, the following thresholds were applied: Low energy 320 counts and high energy 40 counts. Precursors and fragment ions were coupled, using correlations of chromatographic elution profiles in low/high energy traces. Then, peak retention times were aligned across all chromatograms. Peak intensities were normalized to the median distribution of all ions, assuming the majority of signals are unaffected by experimental conditions. The label-free quantification relied on measured peak areas of the three most intense precursor peptides, preferentially unique. Before statistical analysis, data were transformed using inverse hyperbolic sine function. For the protein identification, the Ion Accounting 4.0 (Waters) search algorithm was applied. The reference sequence file was as mentioned above. Workflow parameters for the protein identification searches were: Maximum two possible chymotrypsin miscleavages, a fixed carbamidomethyl cysteine, variable oxidized methionine, and deamidated glutamine. Chymotrypsin was defined as cutting on C-terminus after tyrosine, phenylalanine, tryptophan, leucine, and methionine, but not before proline. The software automatically determined the precursor and peptide fragment mass tolerances. Peptide matching was limited to less than 4% false discovery rate against the randomized database. Identifications were accepted if at least two distinct reliable peptides (score ≥ 5.5, mass accuracy ≤ 15 ppm) matched the protein sequence. The protein grouping feature was then applied to show only hits with unique peptides. We considered as differentially abundant proteins, which satisfied the same strict statistic and effect size criteria as for 2-DE.

To assess the clinical relevance of identified proteins, which differentially accumulated in grain from studied cultivars, we used Allergome (allergome.org), ProPepper (propepper.net), and GluPro [48]. Biological functions and technological quality features of polypeptides were taken primarily from UniProt. Since recently, there was a systematic update of the wheat proteome in UniProt; obsolete accessions were replaced with analogous sequences actual for February 2019 if available. The cumulative coefficient of contrastingly accumulated clinically relevant proteins was calculated by sharing 1.5 points per hit among cultivars. The genotype showing the highest accumulation of a particular allergenic/toxic protein received maximum count while the cultivar with the lowest abundance of the same hit acquired a minimum point.

### 3.6. Data Availability

All data generated or analyzed during this study are included in this published article and its Appendix A. The mass spectrometry proteomics data have been deposited to the ProteomeXchange Consortium via the PRIDE partner repository with dataset identifier PXD012940 [60].

## 4. Conclusions

Human pathologies associated with grain proteins are on the rise, and the only effective treatment is a lifelong gluten-free diet, which is complicated to follow and detrimental to gut health. Therefore, there is a need to improve the quality of life for millions of gluten intolerant patients around the world and prevent new cases. Herein, we showed that the majority of grain proteins, variable among genotypes, belonged to the metabolic group, while detected gliadins were the most health-threatening, due to the highest number/density of epitopes. This group has a somehow lower influence on bread-making quality. Thus, it is promising to investigate promoter polymorphism of gliadin genes further to select germplasm for the breeding of bread wheat cultivars with satisfactory technological properties, yet reduced allergenicity. We discovered the highest accumulation coefficient of allergenic/toxic proteins in landrace Ukrainka. This can be explained by its genetic heterogeneity in contrast to modern genotypes. Among them, particularly Sotnytsia stands out as suitable germplasm for novel less toxic cultivars. Marker peptides from variable proteins can be used in method development for targeted quantification by a triple quadrupole mass spectrometer, offering an accurate high-throughput alternative in both analyte/genotype dimensions, enabling fast and efficient assessment of the medical safety of multiple wheat cultivars.

## Figures and Tables

**Figure 1 ijms-21-03445-f001:**
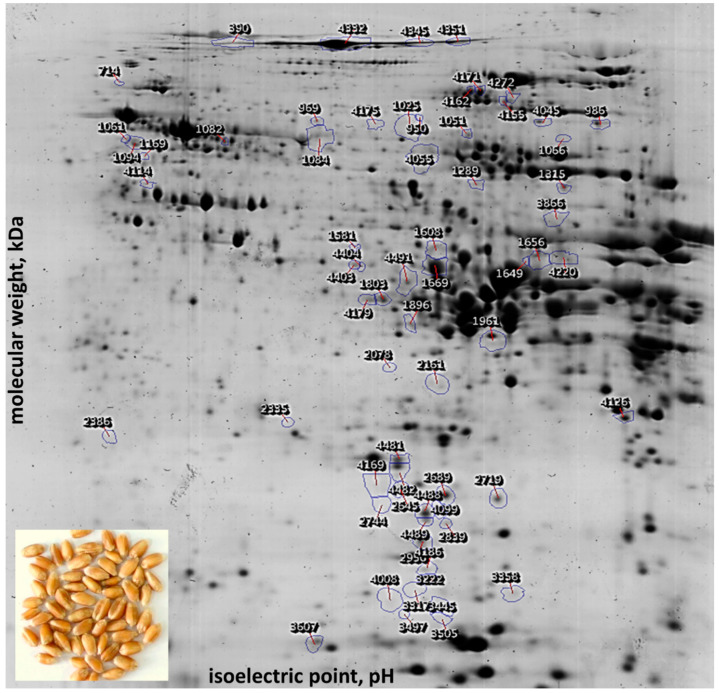
Annotated gel with 66 differentially abundant protein spots among modern and traditional cultivars marked on the image of alignment reference using the SameSpots 5.1 program. In total, 810 analytes extracted from the grain of bread wheat were separated in the 4.0–9.5 pI range and 11–129 kDa molecular mass interval.

**Figure 2 ijms-21-03445-f002:**
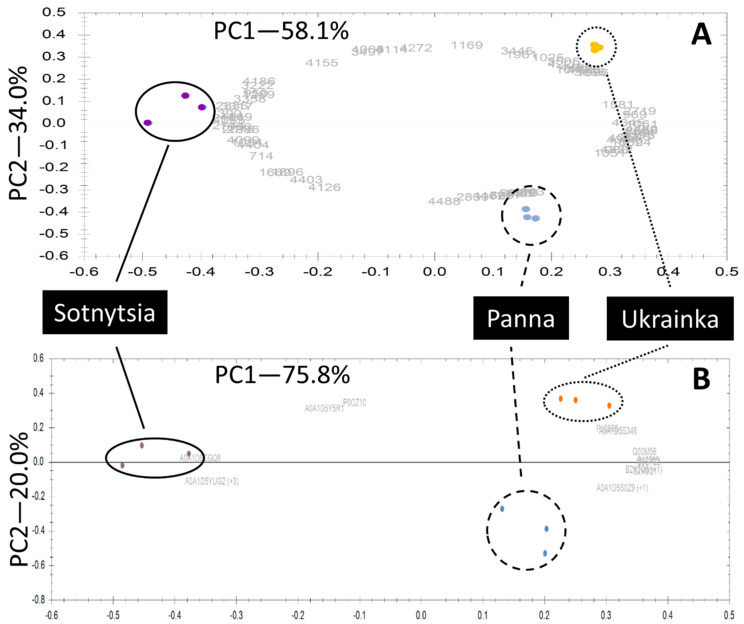
Principal component analysis of differentially abundant proteins showed excellent reproducibility among biological replicates for both analytical approaches and suggested that modern cultivar Sotnytsia is distant: (**A**) 66 gel spots; (**B**) 12 proteins discovered upon direct mass spectrometry quantification. Multidimensional statistics were calculated and visualized using functions integrated into SameSpots 5.1 and Progenesis QI.

**Figure 3 ijms-21-03445-f003:**
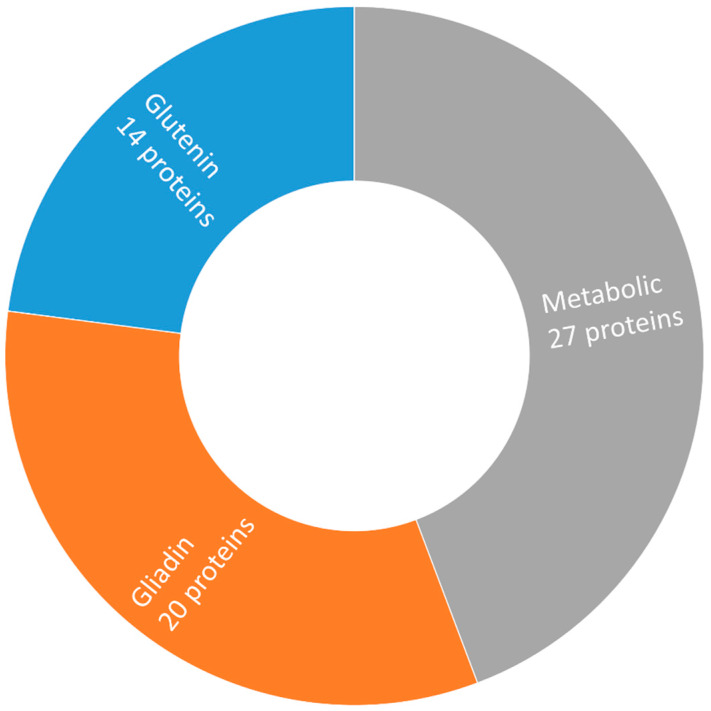
The proportion of grain protein groups differentially accumulated among investigated *Triticum aestivum* L. cultivars revealed the most diverse non-gluten fraction. Information about the functional role of proteins was extracted primarily from UniProt.

**Figure 4 ijms-21-03445-f004:**
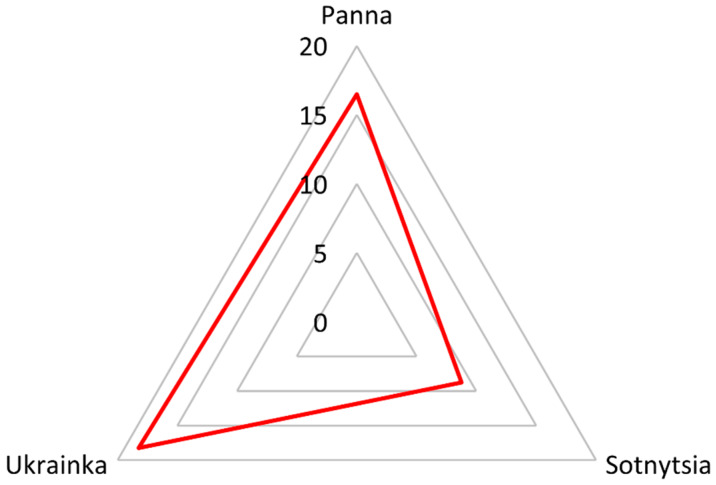
Cumulative coefficient of contrastingly accumulated proteins with harmful epitopes discovered with both experimental approaches among bread wheat genotypes; from a total of 61 identified polypeptides, 29 are of medical concern. Modern cultivar Sotnytsia scored lowest, while landrace Ukrainka accumulated the most allergenic/toxic proteins in grain.

**Table 1 ijms-21-03445-t001:** Identified differentially abundant proteins from *Triticum aestivum* L. cultivars discovered by two-dimensional gel electrophoresis. Bold ratios indicate significant differences in the respective comparison (ratio ≥ 2.5 and Tukey’s test *p* ≤ 0.01). Positive value of proportion—protein is more abundant, negative value—protein is less abundant, NA—not available.

Spot #	UniProt Accession	Protein Name (Genetic Locus)	Function	Allergen or Toxin	Protein Group	Panna/Sotnytsia	Panna/Ukrainka	Sotnytsia/Ukrainka
4126	B2Y2Q6	LMW-B2 glutenin (*Glu-B3-2*)	Nutrient reservoir activity	√	Glutenin	1.07	**2.51**	2.34
1066	P08488	HMW-12 glutenin (*HMW-GS DY*)	Nutrient reservoir activity	√	Glutenin	−2.12	1.19	**2.53**
4114	P08488	HMW-12 glutenin (*HMW-GS DY*)	Nutrient reservoir activity	√	Glutenin	−2.37	**−2.82**	−1.19
390	P08489	HMW-PW212 glutenin	Nutrient reservoir activity	√	Glutenin	**−11.01**	1.08	**11.84**
4045	P10388	HMW-DX5 glutenin (*Glu-1D-1D*)	Nutrient reservoir and starch binding activity	√	Glutenin	**2.65**	1.09	−2.43
4332	P10388	HMW-DX5 glutenin (*Glu-1D-1D*)	Nutrient reservoir and starch binding activity	√	Glutenin	**3.00**	1.34	**−2.24**
4345	P10388	HMW-DX5 glutenin (*Glu-1D-1D*)	Nutrient reservoir and starch binding activity	√	Glutenin	2.47	−1.18	**−2.91**
4351	P10388	HMW-DX5 glutenin (*Glu-1D-1D*)	Nutrient reservoir and starch binding activity	√	Glutenin	**4.14**	1.09	**−3.79**
1896	A1EHE7	γ-gliadin D3 (*Gli1*)	Nutrient reservoir activity	√	Gliadin	−1.32	2.61	**3.45**
1961	A1EHE7	γ-gliadin D3 (*Gli1*)	Nutrient reservoir activity	√	Gliadin	−1.36	**−3.97**	**−2.92**
3866	I0IT62	α/β-gliadin	Nutrient reservoir activity	√	Gliadin	1.40	**−3.72**	**−5.20**
1649	M9TG60	γ-gliadin A1 (*Gli1*)	Nutrient reservoir activity	√	Gliadin	1.24	**−4.40**	**−5.47**
1656	M9TG60	γ-gliadin A1 (*Gli1*)	Nutrient reservoir activity	√	Gliadin	1.37	**−4.09**	**−5.59**
4220	M9TG60	γ-gliadin A1 (*Gli1*)	Nutrient reservoir activity	√	Gliadin	−1.04	**−3.67**	**−3.53**
1669	P04722	α/β-gliadin AII	Nutrient reservoir activity	√	Gliadin	−1.32	**2.57**	**3.37**
4403	P04722	α/β-gliadin AII	Nutrient reservoir activity	√	Gliadin	−1.03	2.49	**2.57**
1803	P21292	γ-gliadin	Nutrient reservoir activity	√	Gliadin	**5.59**	**4.47**	−1.25
4179	P21292	γ-gliadin	Nutrient reservoir activity	√	Gliadin	**3.09**	**2.78**	−1.11
2078	Q94G92	γ-gliadin D4 (*Gli1*)	Nutrient reservoir activity	√	Gliadin	−2.60	1.35	**3.50**
2161	Q94G92	γ-gliadin D4 (*Gli1*)	Nutrient reservoir activity	√	Gliadin	**−3.40**	1.11	**3.78**
4491	A0A1D5T3T7	Similar to δ-gliadin D1, obsolete	Nutrient reservoir activity		Gliadin	1.18	**−4.30**	**−5.08**
2645	A0A3B5YPZ7	Similar to avenin-like protein	Nutrient reservoir activity		Gliadin	**4.00**	1.04	**−3.86**
2689	A0A3B5YPZ7	Similar to avenin-like protein	Nutrient reservoir activity		Gliadin	**6.03**	−1.14	**−6.86**
4169	A0A3B5YPZ7	Similar to avenin-like protein	Nutrient reservoir activity		Gliadin	**−3.16**	1.07	**3.37**
4481	A0A3B5YPZ7	Similar to avenin-like protein	Nutrient reservoir activity		Gliadin	**3.90**	−1.03	**−4.03**
4482	A0A3B5YPZ7	Similar to avenin-like protein	Nutrient reservoir activity		Gliadin	**4.33**	−1.10	**−4.77**
1581	Q41593	Serpin-Z1A (*WZCI*)	Serine protease inhibitor, extracellular	√	Metabolic	1.88	−1.57	**−2.95**
3607	Q43691	Trypsin/α-amylase inhibitor CMX2	Serine protease inhibitor, secreted	√	Metabolic	**2.63**	**2.70**	1.03
4404	A0A1D5US94	Methyltransferase	Protein dimerization activity		Metabolic	−1.87	1.61	**3.00**
1084	A0A1D5YFA7	Similar to β-amylase, obsolete	Hydrolysis of (1->4)-α-D-glucosidic linkages in polysaccharides		Metabolic	−2.55	1.82	**4.64**
2386	A0A1D5ZTV0	Similar to serpin-N3.2, obsolete	Serine protease inhibitor, extracellular		Metabolic	−2.38	1.24	**2.94**
1025	A0A1D6A827	Dehydrin, obsolete	Stress response		Metabolic	−1.21	**−7.08**	**−5.86**
969	A0A1D6D1Q3	Pyrophosphate—fructose 6-phosphate 1-phosphotransferase subunit β (*PFP-β*)	Glycolysis, cytoplasm		Metabolic	2.32	−1.33	**−3.10**
2335	A0A1D6RH21	Similar to dehydroascorbate reductase (*DHAR*), obsolete	Glutathione S-transferase domain		Metabolic	**−5.31**	−1.14	**4.65**
1169	A0A1D6S518	UTP—glucose-1-phosphate uridylyltransferase	Biosynthesis of saccharides, cytoplasm		Metabolic	−1.72	**−2.94**	−1.71
2719	A0A3B5XV32	Cupin domain protein	Nutrient reservoir activity		Metabolic	**2.60**	−1.51	**−3.94**
4272	A0A3B6ILV9	Similar to globulin 3	Nutrient reservoir activity		Metabolic	−2.10	**−2.73**	−1.30
3358	A0A3B6IN56	Similar to alanine-tRNA ligase	Protein synthesis		Metabolic	**−2.93**	−1.22	2.41
1082	A0A3B6KSH4	Similar to β-amylase	Hydrolysis of (1->4)-α-D-glucosidic linkages in polysaccharides		Metabolic	**2.60**	1.10	−2.36
4055	A0A3B6MYZ0	Similar to globulin-1 S allele	Nutrient reservoir activity		Metabolic	**−4.81**	1.19	**5.73**
1051	I6QQ39	Globulin-3A (*Glo-3A*)	Nutrient reservoir activity		Metabolic	**2.66**	1.35	−1.97
4155	I6QQ39	Globulin-3A (*Glo-3A*)	Nutrient reservoir activity		Metabolic	**−3.09**	−2.27	1.37
1315	A0A1D5VMG1	Uncharacterized protein, obsolete	NA		Metabolic	**3.19**	1.03	**−3.11**
950	A0A3B5Z536	Uncharacterized protein	NA		Metabolic	**−5.44**	−1.58	3.44
1289	A0A3B6G0N3	Uncharacterized protein	NA		Metabolic	**−2.63**	−1.28	2.06
4162	A0A3B6JER7	Uncharacterized protein	NA		Metabolic	**4.48**	**5.18**	1.16
4171	A0A3B6JER7	Uncharacterized protein	NA		Metabolic	**3.51**	**3.74**	1.06
4489	A0A3B6LGL1	Uncharacterized protein	NA		Metabolic	**5.72**	**5.05**	−1.13
4008	A0A3B6PFQ8	Uncharacterized protein	NA		Metabolic	**−4.11**	**−4.38**	−1.06

**Table 2 ijms-21-03445-t002:** Differentially abundant proteins among wheat cultivars revealed by direct label-free mass spectrometry quantification. Bold values indicate significant differences in the respective comparison (ratio ≥ 2.5 and Tukey’s test *p* ≤ 0.01). Positive value of proportion—protein is more abundant, negative value—protein is less abundant.

UniProt Accession	Protein Name (Genetic Locus)	Function	Allergen or Toxin	Protein Group	Panna/Sotnytsia	Panna/Ukrainka	Sotnytsia/Ukrainka
Q6SPZ3	LMW-A2 glutenin (*Glu-A3-11*)	Nutrient reservoir activity	√	Glutenin	**6.39**	−1.25	**−7.99**
Q00M56	LMW-D1 glutenin (*Glu-D3-3*)	Nutrient reservoir activity	√	Glutenin	2.31	−1.39	**−3.21**
P10386	LMW-1D1 glutenin (*Glu-D3-2*)	Nutrient reservoir activity	√	Glutenin	1.45	−1.92	**−2.78**
D2DII3	LMW glutenin subunit (*Glu-A3-16*)	Nutrient reservoir activity	√	Glutenin	**4.52**	−1.30	**−5.88**
B2Y2Q6	LMW-B2 glutenin (*Glu-B3-2*)	Nutrient reservoir activity	√	Glutenin	**11.01**	1.07	**−10.24**
B2Y2Q1	LMW glutenin subunit (*Glu-B3-1*)	Nutrient reservoir activity	√	Glutenin	**3.06**	1.07	**−2.86**
A0A1D5S346	Similar to γ-gliadin, obsolete	Nutrient reservoir activity	√	Gliadin	1.48	−1.81	**−2.68**
P0CZ10	Avenin-like a6	Nutrient reservoir activity		Gliadin	−3.06	**−3.38**	−1.11
A0A3B6K9J4	Thaumatin family protein	Multiple disulfide bonds		Metabolic	−1.78	1.44	**2.56**
A0A3B5XV32	Cupin domain protein	Nutrient reservoir activity		Metabolic	**2.91**	1.44	−2.02
A0A1D5Y5R1	Cupin domain protein, obsolete	Nutrient reservoir activity		Metabolic	**−2.74**	**−2.57**	1.07
A0A3B6SDE7	Uncharacterized membrane protein	Integral component of membrane		Metabolic	**−4.88**	1.08	**5.25**

**Table 3 ijms-21-03445-t003:** Detailed characteristics of discovered allergenic or toxic proteins differentially accumulated among bread wheat cultivars. Information about reactive motifs was extracted from dedicated databases—Allergome, ProPepper, and GluPro, NA—not available.

UniProt Accession	Protein Name (Genetic Locus)	Allergome Identifier	Protein Group	# of Epitopes ProPepper	# of Celiac Motifs GluPro	Density of Celiac Motifs GluPro
P08489	HMW-PW212 glutenin	Tri a 26	Glutenin	48	18	0.02
P08488	HMW-12 glutenin (*HMW-GS DY*)	Tri a 26	Glutenin	6	3	< 0.01
P10388	HMW-DX5 glutenin (*Glu-1D-1D*)	Tri a 26	Glutenin	49	5	0.01
B2Y2Q6	LMW-B2 glutenin (*Glu-B3-2*)	Tri a 36	Glutenin	9	5	0.01
Q6SPZ3	LMW-A2 glutenin (*Glu-A3-11*)	Tri a 36	Glutenin	NA	11	0.03
Q00M56	LMW-D1 glutenin (*Glu-D3-3*)	Tri a 36	Glutenin	7	5	0.01
D2DII3	LMW glutenin subunit (*Glu-A3-16*)	Tri a 36	Glutenin	8	NA	NA
B2Y2Q1	LMW glutenin subunit (*Glu-B3-1*)	Tri a 36	Glutenin	12	4	0.01
P10386	LMW-1D1 glutenin (*Glu-D3-2*)	Tri a 36	Glutenin	6	4	0.01
I0IT62	α/β-gliadin	NA	Gliadin	43	13	0.04
P04722	α/β-gliadin AII	NA	Gliadin	127	22	0.08
P21292	γ-gliadin	Tri a 20	Gliadin	59	46	0.15
A1EHE7	γ-gliadin D3 (*Gli1*)	Tri a 20	Gliadin	49	37	0.13
Q94G92	γ-gliadin D4 (*Gli1*)	Tri a 20	Gliadin	40	23	0.08
A0A1D5S346	Similar to γ-gliadin, obsolete	Tri a 20	Gliadin	NA	NA	NA
M9TG60	γ-gliadin A1 (*Gli1*)	Tri a 20	Gliadin	44	31	0.09
Q41593	Serpin-Z1A (*WZCI*)	Tri a 33	Metabolic	NA	NA	NA
Q43691	Trypsin/α-amylase inhibitor CMX2	Tri a CMX	Metabolic	NA	NA	NA

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
