# Peer review of "Comprehensive Comparison of Clinically Relevant Grain Proteins in Modern and Traditional Bread Wheat Cultivars"

_ijms, 2020, doi:10.3390/ijms21103445_

Round 1
Reviewer 1 Report
The paper has an interesting subject and presents new knowledge about diversity of grain proteins in T. aestivum L. However, some parts of ms demands to be rewritten. My suggestions are presented below:
Lines 19, 34, 38, 236: should be Triticum aestivum L.
Line 45: I suggest to add the following citation here: Kuźniar A., Włodarczyk K., Grządziel J., Goraj W., Gałązka A., Wolińska A., 2020. Culture-independent analysis of an endophytic core microbiome in two species of wheat: Triticum aestivum L. (cv. ‘Hondia’) and the first report of microbiota in Triticum spelta L. (cv. ‘Rokosz’). Systematic and Applied Microbiology 43(1), 126025. https://doi.org/10.1016/j.syapm.2019.126025
Introduction is well and logically written.
Line 126: I am not sure if this wheat cultivars description should be placed in Results and Discussion part. From my point of view this material is more suitable to Materials and Methods because in fact you were not presented any of your results here. It is simply cultivars characteristic. Thus is why I suggest to remove this sub-section into Material and Methods.
Lines 146-171: this fragment is also not suitable for Results...It is also a part of methodology. You should first describe your findings and after that to relate them to other literature facts.
In general I suggest you to separate section Results and Discussion, try to describe your results first and after that present confrontation with findings of other studies.
I also did not find any mentioning about Figure 1. Please remember that first you should point about your findings and later present figure. I also suggest to change this figure size in order Figure caption was presented on the same page as figure.
Line 343: Please put Figure 4 caption on the same page as presented figure. You can transfer some text presented before Figure 4 after Figure 4.
Lines 529-533: It is not your conclusions. It is an obvious description of wheat advantages. I suggest to remove this fragment.
Author Response
English language and style
We double-checked the spelling and style of the manuscript and made minor modifications.
Comments and Suggestions for Authors
Thank you very much for the high score of our study, particularly background and design. We improved clarity of Methods and structure of Results and Discussion using your valuable feedback.
The paper has an interesting subject and presents new knowledge about diversity of grain proteins in T. aestivum L. However, some parts of ms demands to be rewritten. My suggestions are presented below:
We carefully addressed your reservations in the revised text, including relocating text, which described genotypes to Materials. Although we feel that combined Results and Discussion allow us to present ideas more clearly and succinctly. Nevertheless, we modified the structure of this section to reflect the expected logic of the presentation. Changes are highlighted in yellow.
Lines 19, 34, 38, 236: should be Triticum aestivum L.
Thank you for spotting a flaw with writing conventional scientific name, we corrected it (L 19, 34, 38, 185, and 313).
Line 45: I suggest to add the following citation here: Kuźniar A., Włodarczyk K., Grządziel J., Goraj W., Gałązka A., Wolińska A., 2020. Culture-independent analysis of an endophytic core microbiome in two species of wheat: Triticum aestivum L. (cv. ‘Hondia’) and the first report of microbiota in Triticum spelta L. (cv. ‘Rokosz’). Systematic and Applied Microbiology 43(1), 126025. https://doi.org/10.1016/j.syapm.2019.126025
We agree that the suggested reference contains valuable information supporting the point in the text; therefore, we cited it while adding to the list of references (L 45 and R 1).
Introduction is well and logically written.
We appreciate your high estimation of the Introduction section.
Line 126: I am not sure if this wheat cultivars description should be placed in Results and Discussion part. From my point of view this material is more suitable to Materials and Methods because in fact you were not presented any of your results here. It is simply cultivars characteristic. Thus is why I suggest to remove this sub-section into Material and Methods.
We respect the referee’s opinion that subsection 2.1. “General Characteristics of genotypes” likely belongs to the Methods section. Now it is part of subsection 3.1. “Plant Material and Protein Extraction” (L 396-413).
Lines 146-171: this fragment is also not suitable for Results...It is also a part of methodology. You should first describe your findings and after that to relate them to other literature facts.
Thank you for you valuable opinion. We agree that the referred paragraph was not in the right place for the logic of the expected structure of a classic manuscript. It discussed the relative strengths and weaknesses of our experimental approach with references to relevant literature. We believe that it would work better still in the Results and Discussion section, yet after primarily results were described. Therefore, the whole section 2.3. “Effective Workflow to Overcome the Analytical Challenges of Grain Proteome Profiling” was relocated (L 264-294).
In general I suggest you to separate section Results and Discussion, try to describe your results first and after that present confrontation with findings of other studies.
We believe that we restored orthodox logic (describing original results and subsequently integrating with knowledge from the literature) by following your suggestion to relocate the description of cultivars into the Methods section. Moreover, the section discussing the strengths of selected and optimized analytical workflow was also placed after mentioning original findings. We feel that the combined Results and Discussion section make the study more succinct, allowing to avoid repetitions.
I also did not find any mentioning about Figure 1. Please remember that first you should point about your findings and later present figure. I also suggest to change this figure size in order Figure caption was presented on the same page as figure.
To use space efficiently, in the first version of the manuscript, Figure 1 was placed before mentioning in the text; this was fixed in the revised version. Now Figure 1 is referenced in the text several times (L 132, 141-145, 152, and 183). We modified it according to the suggestion of the second referee, that allowed to adjust size joining with the caption.
Line 343: Please put Figure 4 caption on the same page as presented figure. You can transfer some text presented before Figure 4 after Figure 4.
Thank you for noticing essential formatting sloppiness, we verified if all figure captions are on the same page, including Figure 4 (L 316-319).
Lines 529-533: It is not your conclusions. It is an obvious description of wheat advantages. I suggest to remove this fragment.
We definitely agree that the first sentence in the pointed passage is indeed common knowledge, thus should be removed. Yet we prefer to retain the remaining two because they reinstate the biological relevance of our findings (L 537-540).
Reviewer 2 Report
In the paper “Comprehensive Comparison of Clinically Relevant Grain Proteins in Modern and Traditional Bread Wheat Cultivars” the authors describe the study of proteomic profiles three Ukrainian wheat cultivars using a combined quantitative approach based on both 2Dgel-MS and LC-MS (label-free quantitation) analysis.
The work is well designed, the results are well commented, and the material and methods section is very detailed.
There are some minor points:
1-authors should insert in supplementary material the list of the identified peptides. This list might be helpful when comparing as, an example, identified HMW-GS sequences. When proteomic techniques are applied to investigate gluten proteins, it is clear that their characterization is challenging. The difficulties arise from the identification of different isoforms belonging to the same group of gluten proteins, differing by point substitutions, insertion or deletion of short sequences in the repetitive domain.
2-Did, the authors performed LC-MS analyses of the peptides extracted from the spot gels and LC-MS of the whole extracts following the same chromatographic conditions (gradient and column?). Please explain in the "Materials and Methods" section.
3-Please, specify if the section "2.5. Both Gluten and Non-Gluten Proteins Contain Allergenic or Toxic Motifs" is performed on data obtained by 2D gel spots analysis or by both gel and LC (label-free).
4-Please deleted Figure 1B and the corresponding sentence in the text. It is quite obvious the complexity of a chromatogram of the digested protein of an extract.
5-Authors should explain the differences between the results of 2D gel approach and LC-MC approached. i.e. the lack of HMW-GS as identified proteins with differences in the abundance.
Author Response
English language and style
We double-checked the spelling and style of the manuscript and made a few adjustments.
Comments and Suggestions for Authors
Thank you very much for the high score of all aspects of our study.
In the paper “Comprehensive Comparison of Clinically Relevant Grain Proteins in Modern and Traditional Bread Wheat Cultivars” the authors describe the study of proteomic profiles three Ukrainian wheat cultivars using a combined quantitative approach based on both 2Dgel-MS and LC-MS (label-free quantitation) analysis.
Thank you for summarizing the main findings of our manuscript. In the revised version, we incorporated your valuable suggestions, particularly revising Figure 1 and adding supplements with matched peptides. The information in some parts of the manuscript was restructured to comply with the suggestion of the first referee, specifically cultivar description in methods; discussion about methodological advances after results). The modified text is highlighted in yellow.
The work is well designed, the results are well commented, and the material and methods section is very detailed. There are some minor points:
We appreciate your feedback helping to streamline the message.
1-authors should insert in supplementary material the list of the identified peptides. This list might be helpful when comparing as, an example, identified HMW-GS sequences. When proteomic techniques are applied to investigate gluten proteins, it is clear that their characterization is challenging. The difficulties arise from the identification of different isoforms belonging to the same group of gluten proteins, differing by point substitutions, insertion or deletion of short sequences in the repetitive domain.
These data with peaklists, matched spectra, and even raw files are available in the dedicated project created in specialized PRIDE archive—PXD012940—with temporary access through username ‘[email protected]’ and password ‘IVZzr0jV’. It will be released immediately if the manuscript is accepted for publication. Nevertheless, we added supplementary materials appropriately referenced in the text (Table S2: Protein and peptide identification details for specific gel spots and Table S4: Peptide identification details for direct liquid chromatography-mass spectrometry quantification) facilitating access for readers (L 138-139, 219-220, and 554-557).
2-Did, the authors performed LC-MS analyses of the peptides extracted from the spot gels and LC-MS of the whole extracts following the same chromatographic conditions (gradient and column?). Please explain in the "Materials and Methods" section.
Thank you for noticing the missing details. In fact, the analysis of gel spots and whole extracts were performed on different equipment nanoAcquity hyphenated with Q-TOF Premier for gel spots and Aquity M-Class coupled with Synapt G2-Si (ion mobility enabled) for whole extracts. Analytical columns and gradient slopes were also different. We amended this information into the revised version (L474 and 476-479).
3-Please, specify if the section "2.5. Both Gluten and Non-Gluten Proteins Contain Allergenic or Toxic Motifs" is performed on data obtained by 2D gel spots analysis or by both gel and LC (label-free).
We started the current section 2.4. „Both Gluten and Non-Gluten Proteins Contain Allergenic or Toxic Motifs“ clearly stipulating that it consolidated 2-DE and LC-MS datasets (L 296).
4-Please deleted Figure 1B and the corresponding sentence in the text. It is quite obvious the complexity of a chromatogram of the digested protein of an extract.
We agree with your judgment, consequently modified Figure 1 and removed the descriptive sentence. The figure caption was also corrected accordingly (L141-145).
5-Authors should explain the differences between the results of 2D gel approach and LC-MC approached. i.e. the lack of HMW-GS as identified proteins with differences in the abundance.
Small overlap between 2-DE and LC-MS results regarding specific differentially abundant proteins was also puzzling for us. Nevertheless, such phenomenon is well documented in the specialized literature (we added relevant citation). In our case, one plausible reason is different analytical depth 810 spots versus 127 quantified proteins, yet the proportion of variable was similar 8% versus 9%. Specifically for HMW glutenins, we assume that 2-DE separated PTMs, contrary to LC-MS. We added a paragraph, which acknowledged the issue and offered explanation (L 257-263, R 37).